# Microwave Irradiation as a Powerful Tool for the Preparation of n-Type Benzotriazole Semiconductors with Applications in Organic Field-Effect Transistors

**DOI:** 10.3390/molecules27144340

**Published:** 2022-07-06

**Authors:** Iván Torres-Moya, Alexandra Harbuzaru, Beatriz Donoso, Pilar Prieto, Rocío Ponce Ortiz, Ángel Díaz-Ortiz

**Affiliations:** 1Department of Inorganic, Organic Chemistry and Biochemistry, Faculty of Science and Chemical Technologies, University of Castilla-La Mancha-IRICA, 13071 Ciudad Real, Spain; beatriz.donoso@uclm.es (B.D.); mariapilar.prieto@uclm.es (P.P.); 2Department of Physical Chemistry, Faculty of Sciences, University of Málaga, Campus of Teatinos s/n, 29071 Malaga, Spain; harbuzaru@uma.es

**Keywords:** microwave irradiation, benzotriazole, OFETs

## Abstract

In this work, as an equivocal proof of the potential of microwave irradiation in organic synthesis, a complex pyrazine-decorated benzotriazole derivative that is challenging to prepare under conventional conditions has been obtained upon microwave irradiation, thus efficiently improving the process and yields, dramatically decreasing the reaction times and resulting in an environmentally friendly synthetic procedure. In addition, this useful derivative could be applied in organic electronics, specifically in organic field-effect transistors (OFETs), exhibiting the highest electron mobilities reported to date for benzotriazole discrete molecules, of around 10^−2^ cm^2^V^−1^s^−1^.

## 1. Introduction

In its early days, microwave radiation was regarded as an unconventional energy source whose application in chemistry seemed almost exotic [1,2]. Nowadays, however, microwave-assisted reactions and chemical processes are well established, and their popularity and applications have increased dramatically, as can be seen from the numerous books or book chapters [3,4,5,6,7,8,9] and reviews [10,11,12,13,14,15] covering this field.

Microwave irradiation produces dielectric heating, as a consequence of the ability of some materials to absorb microwave radiation and convert it into heat by dipolar polarization and ionic conduction mechanisms [16]. Thus, the absorption of microwave energy during chemical reactions causes rapid internal heating due to the direct interaction between electromagnetic radiation and polar substances. As a result, spectacular reaction rate accelerations, cleaner processes and higher yields, which often cannot be reproduced with classical heating, are observed [17,18,19,20,21,22,23,24].

Practically all known chemical processes have been studied under microwave irradiation (substitutions, cyclizations, cycloadditions, C-C cross-couplings, redox processes, etc.), and in most cases, remarkable improvements have been observed [3,4,5,6,7,8,9,10,11,12,13,14,15]. Our research group has been using microwave radiation since 1992 in numerous and varied organic chemical processes in the absence of solvents, such that it is now the most common and sustainable source of energy used in our laboratory [25,26,27,28,29,30,31].

Taking into account all these advantages, organic synthesis under microwave irradiation is of vital importance in a field that has been growing steadily in recent years, namely organic electronics, in which organic compounds can be applied in electronic devices owing to the advantages that they exhibit compared with inorganic compounds, such as lightness, processability, lower cost and easy chemical modulation [32,33,34]. In this sense, the main components in many electronic devices are organic field-effect transistors (OFETs) [35,36,37,38], which are essential building blocks for the next promising generation of cheap and flexible organic circuits.

In the last few years, our research group has been studying the 2*H*-benzo[*d*]1,2,3-triazole (BTz), an electron acceptor moiety that can be easily modified by alkylation at the central nitrogen and/or by introducing different groups in the benzene ring, thus making it an excellent building block with potential applications in different fields [39,40,41,42,43,44]. 

In previous studies, we have described different donor–acceptor–donor (D-π-A-π-D) 2*H*-benzo[*d*]1,2,3-triazole derivatives with applications as OFETs that show p-type semiconductor behavior and moderate mobilities [45]. However, it was necessary to improve these mobility values to allow the use of these systems in electronic devices. For this reason, herein, new structural modifications were carried out in the central benzotriazole core as a “proof-of-concept”, with the aim of improving its electrical behavior in OFETs. Hence, a combination of two dipolar systems in the same structure leads to a multidonor–acceptor molecule with a D-π-A-π-D system in the horizontal axis and a D-A system in the vertical one (Figure 1). In order to perform this kind of orthogonal combination, and to incorporate a new donor group (Donor 2) in the structure, the BTz unit has been expanded with a pyrazine ring, thereby increasing the acceptor character of the central core and resulting in enhanced π-conjugation throughout the structure. Bithiophene was chosen as Donor 1 due to its high donor character and its higher charge transport capacity due to its lower aromatic character with respect to benzene (Figure 1). 

Microwave irradiation has usually been employed in single chemical transformations or in two/three reaction steps of a sequential synthesis. Our challenge in this study was a complete microwave-assisted synthesis, starting from simple commercial reagents and involving eight sequential steps, to give a complex and interesting product (**1**) with potential valuable properties as an organic field-effect transistor (OFET).

## 2. Results

### 2.1. Theoretical Calculations

In order to verify the promising applicability of the chosen dipolar benzotriazole derivative **1** in OFETs, the minimum-energy optimized structures were calculated at the B3LYP/6-31G (d,p) theoretical level (see Figure 2). This compound presents high planarity, thereby favoring the formation of ordered self-assembled films during the manufacture of OFET devices. In addition, it can be seen that the HOMO and LUMO orbitals are confined in different parts of the molecule. The HOMO orbital is mainly spread out along the horizontal axis of the molecule, while the LUMO is found along the benzotriazole core, although a spatial overlap exists between them, thus implying the existence of some intramolecular charge transfer (ICT) within the molecule. In addition, the HOMO–LUMO gap is low (lower than 2.0 eV) and the HOMO and LUMO values are relatively accessible considering the Fermi level of the gold electrodes (−4.8 eV), thus making this system a possible candidate for efficient hole and electron transport in OFETs.

In an effort to gain more insight into the electrical behavior of the compound, the internal reorganization energies were calculated, employing the B3LYP/6-31G (d,p) theory level, through the standard procedure described in the literature [46,47] from the relevant points on the potential energy surfaces. This parameter is related to the energy required to accommodate the electronic and molecular changes upon charge transfer. The reorganization energy corresponding to electron transport for compound **1** (0.14 eV) is lower than that obtained for hole transport (0.20 eV), which may be related to the greater delocalization of the LUMO orbital over the π-conjugated skeleton. As a consequence, this compound may be more efficient as an n-type semiconductor, thus reversing the electrical behavior found for the previous D-π-A-π-D benzotriazole derivatives reported by our research group [45], in which LUMO orbitals were far from the Fermi level for gold and in which only reorganization energies for hole transport were lower than 0.20 eV, thus behaving as p-type semiconductors in all cases. For all these reasons, and encouraged by the different and potentially novel properties of this derivative, we decided to synthesize it, despite the difficulties of the long synthetic route, using microwave radiation during all steps, if possible, to corroborate our commitment to a more sustainable chemistry and to make the process more efficient.

### 2.2. Synthesis

The synthesis of compound **1** is depicted in Figure 1. Thus, microwave irradiation of a mixture of 1-nitro-2-nitrosobenzene (**2**) [48] and 3,5-bis(trifluoromethyl)aniline (**3**) in a small amount of AcOH (1 mL) during 20 min afforded azo derivative **4** in 94% yield. This compound was treated with formamidinesulfinic acid in a mixture of NaOH/*^t^*BuOH and irradiated with microwaves for 30 min to give derivative **6** in 78% isolated yield, which was brominated with Br_2_/AcOH under irradiation at 100 °C for 10 min to afford compound **7** in 80% yield (Figure 1). Under these conditions, we should highlight that it was possible to prepare only the dibrominated product **7**, whereas a mixture of different polybrominated derivatives is obtained under conventional conditions, with the consequent problem of the costly separation of very similar products. It should also be noted that intermediate **7** has previously been prepared by our research group under classical conditions in several steps [48,49], employing higher quantities of solvents, longer reaction times, higher temperatures, and obtaining lower yields than those obtained under microwave irradiation.

Nitration of intermediate **7** with a mixture of HNO_3_/H_2_SO_4_ at 60 °C under microwaves for 10 min gave dinitro derivative **8** in 92% yield [50], which was quantitatively reduced to diamino derivative **9** upon irradiation in the presence of CuNPS, a basic medium, and a small amount of glycerol (1 mL) within 30 min [51]. This part of the synthesis ends with the condensation of **9** with the diketone **10** under microwave irradiation at 100 °C for 30 min to afford the multicyclic compound **11** in very good yield (92%) (Figure 1) [52]. We would like to highlight both the quantitative reduction of the complex dinitro derivative **8** under microwave irradiation and the excellent yields obtained in the nitration and condensation steps: with classical heating, the yield of dinitro derivative **8** did not exceed 60%.

We also prepared the alkynyl derivative **12** by employing a two-step procedure developed by our research group previously (Figure 2). In the first step, 5-bromo-2,2’-bithiophene (**13**) and ethynyltrimethylsilane (**14**) undergo a Sonogashira C-C cross-coupling reaction under microwave irradiation to afford intermediate **15** in excellent yield. This intermediate was desilylated with a mixture of MeOH/THF at room temperature in 30 min, which led to alkynyl derivative **12**. Since this step gives a quantitative yield, we could not improve it by microwave irradiation.

In the final step, a microwave-assisted Sonogashira C-C cross-coupling reaction, published by our research group [49,53,54], between **11** and **12** in the presence of the reusable catalyst Pd-Encat TPP 30, thus making the sustainability of the process more consistent, gave the desired compound **1** within 20 min in good yield (71%). This compound was purified by column chromatography on silica gel employing hexane/ethyl acetate (1:2) as an eluent and gave satisfactory analytical NMR spectroscopic and MS data (NMR spectra can be found in Appendix A). It is important to note that the use of microwave irradiation significantly improves the process, allowing the desired compound to be obtained pure and in good yields. The same Sonogashira cross-coupling reaction was performed previously by our research group with conventional heating, giving the pure product in only 10% yield because of the production of complex reaction mixtures that were difficult to process, probably due to the lack of reactivity of dibromo derivative **11**. Hence, the use of microwave irradiation was an essential tool in this work to obtain the promising and desirable derivative **1**, dramatically improving the efficiency of the process, making it more sustainable, especially in terms of scalability.

Table 1 shows a comparison between conventional conditions and microwave irradiation of all the steps in terms of yields and reaction times—the time-saving when using microwave irradiation (118 h vs. 2.5 h)—thus resulting in a much more practical process. With regard to the performance of the reactions, as can be seen, although the improvement in some individual steps with microwave radiation is not very significant with respect to classical conditions, the overall yield of the whole synthetic process is considerably better when microwave radiation is used (1% vs. 35%).

### 2.3. Photophysical Data

The photophysical spectra of compound **1** were experimentally measured in a DMSO solution at a concentration of 10^−5^ M. The UV–Vis and PL spectra are shown in Figure 3 and the results are summarized in Table 2.

The results showed the influence of the multipolar system on the photophysical properties. Both in the absorption and emission spectra, it can be clearly observed a bathochromic shift in comparison with the previous D-A-D BTZ derivatives previously described by our research group [45,49], due to the higher level of conjugation originated by the interaction of the D-A-D system in the horizontal axis and the D-A in the vertical one, and the lower HOMO–LUMO gap. Furthermore, benzotriazole **1** showed a moderate quantum yield fluorescence of 0.58 due to the presence of the sulfur atom. It is known that the presence of heavy atoms tends to favor intersystem crossover, enhancing phosphorescence and therefore decreasing the fluorescence. 

### 2.4. Electrical Characterization

Once the desired compound **1** had been successfully synthesized and characterized, top-contact/bottom-gate OFETs were fabricated by organic semiconductor vapor deposition on Si/SiO_2_ substrates previously treated with octadecyltrichlorosilane (OTS) or hexamethyldisilazane (HMDS), in order to evaluate the electrical properties of benzotriazole **1**. Finally, the device fabrication was concluded by thermal gold deposition through shadow masks to define the source and drain electrodes. The parameters that usually determine the efficiency of the transistor (field-effect mobility (μ), I_ON_/I_OFF_ ratio and threshold voltage (V_T_)) were extracted from the I-V response plots in the saturation regime, using Equation (1), in which W is the channel width, L the channel length, C the capacitance per unit area of the insulator layer, and V_G_ the gate voltage [55,56].
(I_D_)_sat_ = (W/2L)μC(V_G_ − V_T_)^2^
(1)

The electrical parameters of benzotriazole **1** are summarized in Table 3. It should be pointed out that these results refer to the films treated with HMDS because they provide the best electrical characteristics. The parameters were calculated at saturation (V_D_ = 100 V) for electron transport. Some representative output and transfer plots are shown in Figure 4.

The results summarized in Table 3 show the dramatic change in electrical behavior for this derivative in comparison with the D-π-A-π-D benzotriazoles described previously by our research group [45], summarized in Appendix A, reversing the charge transport polarity and also significantly increasing the mobility by several orders of magnitude (10^−2^ cm^2^V^−1^s^−1^ vs. 10^−4^ or 10^−5^ cm^2^V^−1^s^−1^). To the best of our knowledge, these are the highest electron mobilities reported for this kind of benzotriazole derivative. In the literature, we can find only some examples about polymers in which the benzotriazole unit appears in combination with other heterocycles, including thiophene and benzothiadiazole, among others, with mobilities ranging from 10^−5^ to 10^−2^ cm^2^V^−1^s^−1^, in which only p-type mobilities were registered [57]. In contrast, in our work, we are not dealing with polymers, but with discrete molecules, in which we have achieved high electron mobilities. As far as we know, derivatives of benzotriazole, with such high electron mobilities, have not been previously described. Furthermore, it should be noted that the results agree with the theoretical calculations performed previously, indicating the usefulness of computational studies as a predictive tool. In fact, the obtained theoretical results predicted more efficient electron transport, considering both the values of the molecular frontier orbitals and of the internal reorganization energies. 

In order to further deepen the results obtained, X-ray diffraction (XRD) studies were carried out. The XRD pattern of the semiconducting films prepared for benzotriazole **1** under the optimal device fabrication conditions is shown in Figure 5.

The XRD pattern shows a sharp peak around 2θ = 8°, confirming some degree of crystallinity, supported by the completely planar structure of this derivative, previously confirmed by theoretical calculations. The presence of ordered aggregates within the thin film can further support the good electrical performance of this benzotriazole derivative.

In summary, this proof of concept opens the door to new rational modifications for the synthesis of similar derivatives, with improved properties as semiconductors in OFETs, once the microwave irradiation synthetic procedure has been optimized. 

## 3. Materials and Methods

### 3.1. General Techniques

The reagents needed for the synthetic procedures were employed without any previous purification. Air-sensitive reactions were performed, employing an argon atmosphere. Microwave irradiations were carried out in a Discover^®^ (CEM, Matthews, NC, Canada) focused microwave reactor with a maximum power of 150W. Measurements and temperature control were carried out thanks to an infrared reader, and the parameters were recorded using the software designed by CEM. Flash chromatography was performed using silica gel (Merck, Kieselgel 60, 230–240 mesh, Merck, Darmstadt, Germany). Analytical thin layer chromatography (TLC) was performed using aluminum-coated Merck Kieselgel 60 F254 plates (Merck, Darmstadt, Germany). ^1^H- and ^13^C-NMR spectra were recorded in a Varian Unity 500 (^1^H 500 MHz; ^13^C 125 MHz) spectrometer (Varian, Palo Alto, CA, USA) at 298 K, employing partially deuterated solvents as an internal reference. Coupling constants (*J*) are described in Hz and chemical shifts (δ) in ppm. Multiplicities are denoted as: s = singlet, d = doublet, t = triplet, m = multiplet, br = broad.

MALDI-TOF mass spectra were acquired in a Bruker Autoflex II TOF/TOF spectrometer (Bruker, Billerica, MA, USA), employing dithranol as the matrix. Samples co-crystallized with the matrix on the probe were ionized with a nitrogen laser pulse (337 nm) and accelerated at 20 kV with time-delayed extraction before entering the time-of-flight mass spectrometer. Matrix (10 mg/mL) and sample (1 mg/mL) were dissolved in different vials in methanol and then mixed in different ratios of matrix/sample from 100:1 to 50:1. As is typical, a 5 μL mixture of matrix and sample was applied to a MALDI-TOF MS probe and air-dried. MALDI-TOF MS in positive reflector mode was used for all samples. External calibration was performed using Peptide Calibration Standard II (mass range: 700–3200 Da) from Care (Bruker, Billerica, MA, USA). The applied peak (*m*/*z* determination) detection method was the threshold centroid at 50% height of the peak maximum.

Theoretical calculations were carried out with Gaussian 16 software (Gaussian Inc., Wallingford, CT, USA) [58], available in the High-Performance Computing Service of UCLM. The calculations were performed within the Density Functional Theory (DFT) framework [59]. Geometry optimizations were calculated with the B3LYP functional [60] and the medium-sized 6-31G (d,p) basis set. All geometrical parameters were allowed to vary independently. Harmonic frequency calculations were performed on the resulting optimized geometries and no imaginary frequencies were observed, a fact that ensured the identification of the global minimum energy.

The intramolecular reorganization energy (λ) consists of two terms related to the geometry relaxation energies upon moving from the neutral state to a charged molecular state and vice versa, and it can be defined by the following Equation (2):λ = λ_rel_^(1)^ + λ_rel_^(2)^(2)
where λ_rel_^(1)^ and λ_rel_^(2)^ were computed directly from the adiabatic potential energy surfaces as:λ_rel_^(1)^ = E^(1)^(N) − E^(0)^(N)(3)
λ_rel_^(2)^ = E^(1)^(I) − E^(0)^(I)(4)
where E^(0)^(N) and E^(0)^(I) are the ground-state energy of the neutral and the radical cation or anion molecular state, respectively; E^(1)^(N) is the energy of the neutral molecule at the optimal ion geometry, and E^(1)^(I) is the energy of the ionized state at the optimal geometry of the neutral molecule.

All top-contact/bottom-gate OFETs were manufactured using benzotriazole **1** as the active layer. Gate dielectrics (p-doped Si wafers with 300 nm thermally grown SiO_2_ dielectric layers) were functionalized with either a hexamethyldisilazane (HMDS) or octadecyl-trichlorosilane (OTS) self-assembled monolayer. The capacitance of the 300 nm SiO_2_ gate insulator was 10 nF cm^–2^. The wafers were cleaned by immersing them twice, for 30 s each time, in EtOH with sonication, drying with a stream of N_2_ and treating with UV–ozone for 10 min before surface functionalization. The cleaned silicon wafers were then treated with hexamethyldisilazane (HMDS) by exposing them to HMDS vapor at room temperature in a closed air-free container under argon for a week, or they were treated with octadecyl-trichlorosilane (OTS) by immersion in a 3.0 mM humidity-exposed solution of OTS in hexane for 1 h, following a previously reported procedure [61]. After OTS deposition, the substrates were sonicated with hexane, acetone, and, finally, with ethanol, and dried with an N_2_ stream. The semiconductors were then vapor-deposited on preheated substrates, followed by gold sublimation through shadow masks to define source and drain electrodes.

### 3.2. Experimental Data

Synthesis of 10-([2,2′-bithiophen]-4-ylethynyl)-14-([2,2′-bithiophen]-5-ylethynyl)-12-(3,5-bis(trifluoromethyl)phenyl)-12*H*-dibenzo[a,c][1,2,3]triazolo[4,5-i]phenazine (**1**)

A mixture of dibromobenzotriazole **11** (0.100 g, 0.146 mmol), the acetylene derivative **12** (0.055 g, 0.292 mmol), DBU (0.044 g, 0.292 mmol), CuI (0.001 g, 0.007 mmol), and Pd-EncatTM TPP30 (0.013 g, 0.005 mmol) was added to a dried microwave vessel under argon. CH_3_CN (1 mL) was then added, and the vessel was closed and irradiated under microwave irradiation at 130 °C for 20 min. The reaction mixture was purified by chromatography, employing a mixture of hexane/ethyl acetate (1:2) as eluent to give a dark red solid **1** (0.092 g, 71%), which was successfully characterized. M.p.: >400 °C. ^1^H-NMR (δ, DMSO, 500 MHz): 8.55 (s, 2H, *o*-N-Ph), 8.31–8.30 (d, *J* = 8.3 Hz, 2H, H-Ph), 8.18 (s, 1H, *p*-N-Ph), 8.04–8.02 (d, *J* = 8.3 Hz, 2H, H-Ph), 7.81–7.78 (t, *J* = 8.3 Hz, 2H, H-Ph), 7.56–7.53 (m, 4H, H-Ph + H-thiophene), 7.30–7.29 (d, *J* = 5 Hz, 2H, H-thiophene), 7.21–7.20 (d, *J* = 5 Hz, 2H, H-thiophene), 7.14–7.10 (d, *J* = 5 Hz, 2H, H-thiophene), 7.09–7.08 (t, 2H, H-thiophene). ^13^C-NMR (δ, DMSO, 125 MHz): 147.1, 145.8, 142.2, 140.9, 140.1, 132.3, 132.2, 132.1, 131.1, 129.4, 128.7, 127.8, 126.9, 126.3, 125.9, 124.1, 123.3, 122.6, 121.8, 112.31, 111.2, 105.9, 84.1, 74.8. MS (*m*/*z*), calculated for (C_48_H_21_F_6_N_5_S_4_) 909.06 g/mol, found 909.54 g/mol.

The rest of the procedures for the other intermediate compounds can be found in the Appendix A.

## 4. Conclusions

A two-dipolar multidonor–acceptor benzotriazole derivative has been synthesized in several reaction steps, employing microwave radiation as an energy source in all steps in which heating was necessary. Thereby, considerably reduced reaction times and improved yields were observed, making the process much more sustainable and efficient in terms of scalability. This compound has been tested in OFETs, showing excellent n-type semiconductor character properties with the highest electron mobilities (around 10**^−^**^2^ cm^2^V**^−^**^1^s**^−^**^1^) reported to date for this kind of derivative. It should be noted that the structural modifications carried out on the benzotriazole derivative notably change the electrical behaviour of previously reported D-A-D derivatives from p-type to n-type semiconductors. This successful “proof of concept” opens the door to future modifications in order to further improve the applicability in OFETs irrespective of the complexity of the process, a problem that has been solved by the use of microwave irradiation.

## Data Availability

Not applicable.

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
