# Peer review of "Microwave Irradiation as a Powerful Tool for the Preparation of n-Type Benzotriazole Semiconductors with Applications in Organic Field-Effect Transistors"

_molecules, 2022, doi:10.3390/molecules27144340_

Round 1
Reviewer 1 Report
Microwave irradiation is an important synthetic tools for organic chemistry. Torres-Moya et al designed a n-type benzotriazole semiconductor and synthesized with the assistance of microwave. However, this paper is still lacking of basic characterizations for the synthesized compound such as UV-vis (solution and film), PL, DSC and XRD. All of these measurements are of vital importance for the new compound and make the paper more readable to audiences. Furthermore, the transistor part should also be significantly improved. i) in line 192, authors mentioned the use of VD=-100 V for holes. I am confused about this because the compound synthesized in this work is a n-type one. ii) in Fig.3b, the log scale in y-axis is recommended. iii) the hysteresis of transfer curve is lacking. iv) why the on/off ratio is so low? On the basis of above considerations, this work cannot be accepted for publication before major revisions.
Author Response
Dear Editor:
First of all, thanks a lot for your e-mail regarding our manuscript entitled: “Microwave Irradiation as a Powerful Tool for the Preparation of n-type Benzotriazole Semiconductors with Applications in Organic Field-Effect Transistors”. Furthermore, we would like to thank the reviewers for their useful suggestions.
We send you a revised version of our manuscript. We have included all the suggestions made by the reviewers and we have responded all their questions. We hope that this revised version is now suitable for publication.
Reviewer 1
Microwave irradiation is an important synthetic tools for organic chemistry. Torres-Moya et al designed a n-type benzotriazole semiconductor and synthesized with the assistance of microwave. However, this paper is still lacking of basic characterizations for the synthesized compound such as UV-vis (solution and film), PL, DSC and XRD. All of these measurements are of vital importance for the new compound and make the paper more readable to audiences.
Response: We are very grateful for these suggestions in order to improve the quality of our manuscript. In this sense, UV-vis and PL in solution, and XRD studies have been added in the new version. We would have liked to have been able to perform UV-vis and PL studies for the film, but unfortunately, we do not have the necessary equipment to perform these measurements. On the other hand, DSC studies were not performed because the melting point of the benzotriazole derivative 1 is higher than 400ºC, suggesting a great thermal stability, and we did not have problems for the sublimation of this derivative when we fabricated the OFET devices. For this reason, we do not consider essential to perform DSC studies.
Furthermore, the transistor part should also be significantly improved.
i)in line 192, authors mentioned the use of VD=-100 V for holes. I am confused about this because the compound synthesized in this work is a n-type one.
Response: We would like to thank the reviewer his comment and the opportunity to clarify this aspect. Indeed, this synthesized compound showed only n-type mobility, hence we agree with the reviewer that the information regarding p-type transport should be removed. We only mentioned the voltage used for holes, because we also tested the devices applying negative voltages but without having any output. Thus, in order to avoid misunderstandings, we have rewritten lines 192 and 193 mentioning only the electron transport characteristics, which is more appropriate and clearer for the reader.
ii) in Fig.3b, the log scale in y-axis is recommended.
Response: We completely agree with this suggestion. In the new version, Figure 3b has been changed by a new one in which y-axis is in log scale.
iii) the hysteresis of transfer curve is lacking.
Response: We agree that it would be very interesting to show the hysteresis of transfer curve in Figure 3b, but unfortunately we did not calculate it when we performed the OFETs measurements.
iv) why the on/off ratio is so low?
Response: The low on/off ratio of aprox. 103 can be due to leakage current effects. In fact, to measure transistor parameters we did not pattern individual OFETs, which can be the reason behind the high off currents and therefore, the moderately low on/off ratios.
Reviewer 2 Report
The manuscript reported the preparation of n-type Benzotriazole Semiconductors by Microwave Irradiation and applied in OFETs. Compared with traditional ones, Microwave Irradiation improves the process and yields. Moreover, electron mobilities of around 10-2 cm2V-1s-1 was reported. There are a few critical points noted below that should be addressed to improve the manuscript. Therefore, I would not recommend to publish it at the current version.
1. In Figure 2, Values of HOMO and LUMO should be -4.9 and -3.23 eV not -4,9 and -3,23 eV.
2. LUMO is -3.23 eV, how could authors claim it to be close to Fermi level of the gold electrode (-4.8 eV)?
3. Some words overlap for Scheme 1 and 2.
4. Which curve is -20 mV in Figure 3a? Blue, black or others?
5. What frequency and intensity of Microwave Irradiation are applied in this study?
6. “To the best of our knowledge, these are the highest electron mobilities reported for this kind of benzotriazole derivatives”, authors claimed the highest electron mobilities for this kind of benzotriazole derivatives were achieved, therefore it would be better to summarize other reported electron mobilities.
Author Response
Dear Editor:
First of all, thanks a lot for your e-mail regarding our manuscript entitled: “Microwave Irradiation as a Powerful Tool for the Preparation of n-type Benzotriazole Semiconductors with Applications in Organic Field-Effect Transistors”. Furthermore, we would like to thank the reviewers for their useful suggestions.
We send you a revised version of our manuscript. We have included all the suggestions made by the reviewers and we have responded all their questions. We hope that this revised version is now suitable for publication.
Reviewer 2
The manuscript reported the preparation of n-type Benzotriazole Semiconductors by Microwave Irradiation and applied in OFETs. Compared with traditional ones, Microwave Irradiation improves the process and yields. Moreover, electron mobilities of around 10-2 cm2V-1s-1 was reported. There are a few critical points noted below that should be addressed to improve the manuscript.
- In Figure 2, Values of HOMO and LUMO should be -4.9 and -3.23 eV not -4,9 and -3,23 eV.
Response: According to this suggestion, we have changed Figure 2 by a new one, in which, we have modified the values of HOMO, LUMO and HOMO-LUMO gap.
- LUMO is -3.23 eV, how could authors claim it to be close to Fermi level of the gold electrode (-4.8 eV)?
Response: We would like to thank the reviewer for the remark. We understand that the original claim was not accurate. In the new version we have modified the sentence. It now reads “the HOMO and LUMO values are relatively accessible considering the Fermi level of the gold electrodes (-4.8 eV), thus making this system a possible candidate for efficient hole and electron transport in OFETs.”
For clarification, what we wanted to state in the original sentence was that many n-type semiconductors have LUMO levels comparable to the one registered here, for this reason, we considered it to be close enough to the Fermi level of the gold electrode for effective electron transport to occur.
- Some words overlap for Scheme 1 and 2.
Response: We have changed in Scheme 1 the word “procedure” by “route” to avoid overlap in Scheme 1 and Scheme 2.
- Which curve is -20 mV in Figure 3a? Blue, black or others?
Response: The curve corresponding to -20 V in Figure 3a is the green one. In order to clarify this aspect, we have changed Figure 3a, now indicating the voltages corresponding to each curve.
- What frequency and intensity of Microwave Irradiation are applied in this study?
Response: In this study, for all the synthesis described above, we have worked with a maximum power of 150W for the microwave irradiation in all the synthesis described.
- “To the best of our knowledge, these are the highest electron mobilities reported for this kind of benzotriazole derivatives”, authors claimed the highest electron mobilities for this kind of benzotriazole derivatives were achieved, therefore it would be better to summarize other reported electron mobilities.
Response: We would like to thank the reviewer the opportunity to clarify this sentence. In the new version, these new sentences are included “In literature, we can find some examples about polymers in which the benzotriazole unit appears in combination with other heterocycles, including thiophene and benzothiadiazole among others, with mobilities ranging from 10-5 to 10-2 cm2V-1s-1 (for example, Journal of Industrial and Engineering Chemistry, 2020, 86, 150–157). However, in that work only p-type mobilities were registered. In contrast, in our work, we are not dealing with polymers, but with discrete molecules, in which we have achieved high electron mobilities. As far as we know, derivatives of benzotriazole, with this high electron mobilities had not been previously described”